# Human-sized magnetic particle imaging for brain applications

M. Graeser [1,2], F. Thieben [1,2], P. Szwargulski [1,2], F. Werner[1,2], N. Gdaniec [1,2], M. Boberg [1,2], F. Griese [1,2], M. Möddel [1,2], P. Ludewig[3], D. van de Ven[4], O.M. Weber [5], O. Woywode[6], B. Gleich[7] & T. Knopp [1,2]

Determining the brain perfusion is an important task for diagnosis of vascular diseases such as occlusions and intracerebral haemorrhage. Even after successful diagnosis, there is a high risk of restenosis or rebleeding such that patients need intense attention in the days after treatment. Within this work, we present a diagnostic tomographic imager that allows access to brain perfusion quantitatively in short intervals. The device is based on the magnetic particle imaging technology and is designed for human scale. It is highly sensitive and allows the detection of an iron concentration of 263 $pmol_{Fe}\,ml^{-1}$, which is one of the lowest iron concentrations imaged by MPI so far. The imager is self-shielded and can be used in unshielded environments such as intensive care units. In combination with the low technical requirements this opens up a variety of medical applications and would allow monitoring of stroke on intensive care units.

[1] Section for Biomedical Imaging, University Medical Center Hamburg-Eppendorf, 20251 Hamburg, Germany. [2] Institute for Biomedical Imaging, Hamburg University of Technology, 21073 Hamburg, Germany. [3] Department of Neurology, University Medical Center Hamburg-Eppendorf, 20251 Hamburg, Germany. [4] Sensing and Inspection Technologies GmbH, 50354 Huerth, Germany. [5] Philips GmbH Market DACH, 22335 Hamburg, Germany. [6] Imaging Components, Philips Medical Systems DMC GmbH, 22335 Hamburg, Germany. [7] Research Laboratories, Philips GmbH Innovative Technologies, 22335 Hamburg, Germany. Correspondence and requests for materials should be addressed to M.G. (email: matthias.graeser@tuhh.de)

Neurovascular diseases such as ischemic stroke, intracranial hemorrhage (ICH), and traumatic brain injury (TBI) are some of the most severe conditions requiring immediate medical attention and diligent monitoring after treatment. With 17 million cases per year worldwide, stroke is the second most common single cause of death (with approximately 5.3 million deaths, second only to ischemic heart disease) and one of the leading causes of disability[1]. Since about 2 million neurons die every minute after acute stroke, time is a very critical factor for a successful treatment. In turn, the requirements for any diagnostic imaging technique are high. The method has to be fast, easily accessible, and convenient to use.

The common clinical imaging techniques used for the brain are computed tomography (CT) and magnetic resonance imaging (MRI). Both techniques have their pros and cons. CT has a high spatial resolution, but it exposes the patient to radiation, and in turn should not be used for recurring monitoring applications after stroke treatment. In addition, the usage of iodine based contrast agents is a contraindication for patients with kidney disease. MRI mainly suffers from a limited accessibility for intensive care unit (ICU) patients and relatively long examination times. As both techniques require dedicated rooms, no modality is currently available for continuous monitoring of the brain perfusion within the ICU. In consequence, this leads to a heavy workload of medical staff that has to control the brain function by motor tests, paralysis checks and ocular reaction. Imaging of the brain is currently only done in a 24 h interval or if a worsening of the patients status is observed. However, if the patient is put in artificial coma and has to be ventilated, the transport is a complex and risky process.

A very promising method for brain imaging is magnetic particle imaging (MPI) that was introduced in 2005 by Gleich and Weizenecker[2]. MPI uses an inhomogeneous magnetic field (selection-field), a temporally varying magnetic field in the kHz-range (drive-field), and a homogeneous field (focus-field) to increase the spatial coverage. MPI is a tracer-based imaging method and measures the concentration of super-paramagnetic iron-oxide (SPIO) nanoparticles. The particles are usually administered intravenously and allow imaging of the vascular system and organ perfusion. In MRI such contrast agents are used for various applications including the detection of lymph node metastasis[3], imaging of liver tissue[4] and imaging of intra-abdominal lesions within the bowel[5]. Within a preclinical setting MPI has already proven to be very fast (with more than 46 volumes per second[6]), to provide good spatial resolution below one millimeter[2], to be highly sensitive with a detection limit of about 5 ng of iron[7,8] and allow for flexible coil designs[9]. Application-wise, MPI has proven to be capable of detecting ischemic stroke with high sensitivity and high temporal resolution in a mouse model[10]. It has also been shown to be suitable for imaging gut bleeding[11], lung perfusion[12], labeled stem cells[13], cerebral aneurysms[14], cancer[15], and cerebral blood volume[16]. In addition, MPI has been shown to be a very useful tool in interventional applications[17–19] where it can even be used for catheter steering[20].

The major barrier for the transition from a preclinical setting to clinical use and the application of MPI on a human-scale has been the lack of an imaging device with sufficient bore size. Most systems described so far in the published literature have a free bore diameter between 3 cm and 12 cm[21] and thus can only accommodate mice and rats. An attempt at constructing a human torso imager was reported in[22]. The authors found that building a human-sized system with a target resolution of 1 mm will be a challenging task, requiring sophisticated and expensive highest-end hardware. One important challenge when upscaling small imaging systems to the human-scale is the large increase of technical effort. I.e. systems featuring high gradient strength of $2\,\mathrm{Tm}^{-1}\,\mu_0^{-1}$ require high current densities in the field generating coils. As a result such systems have high electrical power dissipation, which needs an appropriate cooling system integrated into the building. Due to these technical challenges, the proof that MPI is clinically useful and sensible is still pending. Nevertheless, simulation studies on the design of a functional MPI brain imager proved promising capabilities for human scale systems[23].

Within this work, we take an alternative approach and present a human-sized MPI system that has low technical requirements designed for flexible and fast deployment in the clinical environment. The system is tailored for brain applications and has a bore size of 19 cm to 25 cm fitting an adult's head. We keep the technical requirements as low as possible while still fulfilling the clinical requirements for a monitoring device for cerebral diseases. The scanner allows a small footprint in the ICU, which is necessary where already several life-supporting and monitoring devices are used. The system can be mounted bedside within the stroke or intensive care unit and it can be used for regular monitoring of the neurovascular status. Based on quantitative perfusion maps and objective criteria, time consuming and risky CT and MRI scans, as well as patient transport efforts are avoided, reducing the risk for patients and the workload for medical staff. The device is evaluated in technical experiments, and the suitability for neuro-imaging is evaluated in static, as well as dynamic experiments using clinically approved tracer concentrations.

## Results

**System overview**. An overview of the developed brain scanner is given in Fig. 1. It consists of a drive-field coil for signal excitation in $x$-direction (caudal-cranial), a dynamic selection-field generator for spatial encoding, and a robot system for slice selection. All components are designed in a way that no additional cooling of the system or shielding of the room is required. For the application scenario of stroke and cerebral hemorrhage detection a spatial resolution of about 1 cm would be sufficient. This allows the system to be designed with an reduced gradient strength of $0.2\,\mathrm{Tm}^{-1}\,\mu_0^{-1}$ in $y$-direction, which is about 10% of the gradient strength of preclinical systems[24]. As encoding scheme a field-free-point (FFP) configuration is used. Thus, in $x$-direction and $z$-direction the gradient provides only half of the strength in comparison to the $y$-direction.

The amplitude of the magnetic field generated by the drive-field coil is chosen to be $6\,\mathrm{mT}\,\mu_0^{-1}$, which can be implemented without any form of active cooling. One potential risk of the drive-field amplitude is the stimulation peripheral of nerves (PNS). According to refs. [25,26], the limit for torso applications is about $3\,\mathrm{mT}\,\mu_0^{-1}$. As the PNS limit scales linearly with the cross section facing the magnetic fields applying $6\,\mathrm{mT}\,\mu_0^{-1}$ for head applications should be safe based on the sizes of an average human.

Using a static selection-field, the drive-field would sample only a single line in $x$-direction. To sample a 2D slice, we use a dynamic selection-field that moves the FFP slowly in $y$-direction with a repetition time of 0.5 s. In combination with the drive-field the FFP moves along a Cartesian sampling trajectory and covers a field-of-view (FOV) of about $100 \times 140\,\mathrm{mm}^2$ in the horizontal plane. For slice selection in $z$-direction, the selection-field coils are mounted on a soft-iron yoke that can be mechanically rotated using a servo motor and additionally moved in $x$-direction using a linear motor. An in-depth discussion on the hardware setup can be found in the 'Methods' section.

**Sensitivity study**. To determine the sensitivity of the imager we used a protocol developed in ref. [7]. First, a dilution series of the

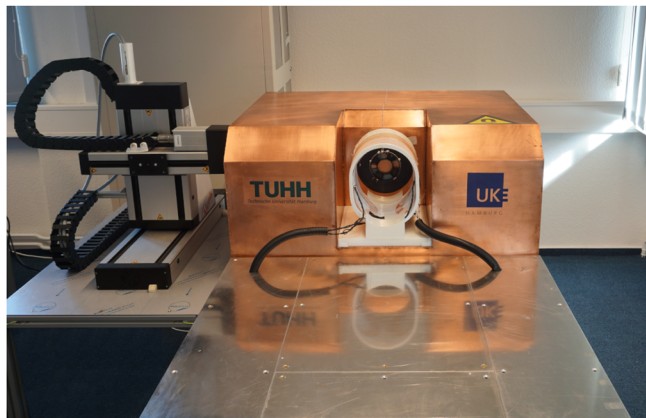

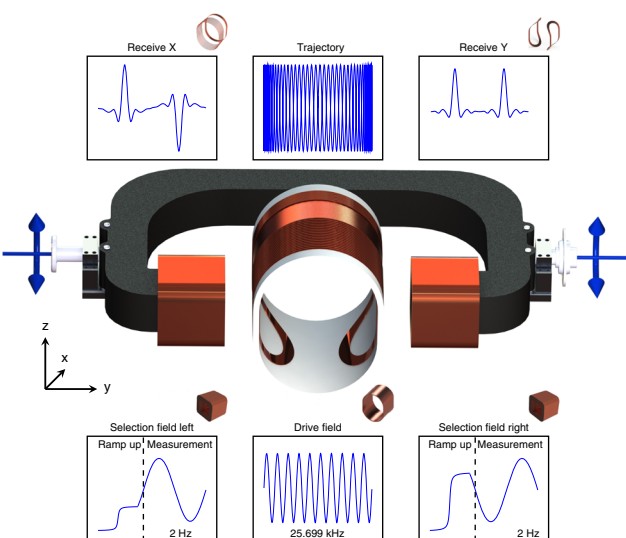

**Fig. 1** Implementation and concept of the presented system. Picture of the developed MPI head scanner. The system-calibration robot is placed to the left of the imager (top). Scanner concept (bottom). The central coil is used as a drive-field generator with a frequency of 25.699 kHz and an amplitude of 6 mT $\mu_0^{-1}$. The cross section is formed by two half circles connected by straight parts forming an ellipsoid-like shape. The selection-field coils are driven by sinusoidal currents with opposite signs superimposed by an offset current. This generates an FFP oscillating on the $y$-axis in-between the selection-field coils. The superposition of the drive-field and the selection-field generates a Cartesian like FFP trajectory covering an FOV within the $xy$-plane of $100 \times 140$ mm$^2$. The particle response is received by dedicated coils in $x$-direction and $y$-direction

tracer Perimag (micromod) was prepared with 50 µl samples and varying iron mass between 2 µg$_{Fe}$ and 512 µg$_{Fe}$. For each iron mass, the sample was moved to three different positions within the FOV, so that a distinction could be made between the sample signal and reconstruction artifacts. All images were recorded using a 2D imaging sequence. Results of the sensitivity study are shown in Fig. 2. It was possible to detect the sample without artifacts starting from 512 µg$_{Fe}$ down to 8 µg$_{Fe}$. For 2 µg$_{Fe}$ several artifacts appeared, but the movement from the upper left to the lower right corner was still visible. For the control measurement, no moving sample could be observed. Therefore, the detection limit of the scanner is about 2 µg iron.

To translate the iron mass into a concentration, we prepared a concentration series and used an ellipsoid (half-axes 40 mm, 40 mm and 20 mm $\hat{=}$ 134 ml) filled with different SPIO concentrations. The ellipsoid was filled with 1 µg to 20 µg iron in 1 µg steps leading to concentrations varying between 7.94 ng$_{Fe}$ ml$^{-1}$ and 147 ng$_{Fe}$ ml$^{-1}$. To determine the sensitivity limit for the concentration series we used the same experimental protocol as the iron mass study. As can be seen in Fig. 2, it was possible to detect the sample for concentrations starting at 147 ng$_{Fe}$ ml$^{-1}$ down to 14.7 ng$_{Fe}$ ml$^{-1}$. For the control experiment no movement of the sample could be detected. Thus, the detection limit in terms of concentration is about 14.7 ng$_{Fe}$ ml$^{-1}$ (263 pmol$_{Fe}$ ml$^{-1}$, 2 µg$_{Fe}$ total), which is one of the lowest iron concentrations imaged by MPI so far.

**Spatial resolution study**. The targeted image resolution of the system was designed to be better than 10 mm to provide sufficient image quality for imaging brain perfusion. To study the spatial resolution a phantom was designed carrying two sample chambers each filled with 250 µl Perimag at a concentration of $c_{Fe} = 8.5$ mg$_{Fe}$ ml$^{-1}$. The distance between the two samples was then varied between 5 mm and 9.5 mm for the $x$-direction and $y$-direction and between 24 mm and 40 mm for the $z$-direction (edge to edge). The orientation of the two dots can be changed such that the resolution limit can be determined in all directions. All images were recorded with 2D imaging sequences.

The reconstructed images, as well as profiles through the sample position are shown in Fig. 3. The two dots with a separation of 9.5 mm could be resolved in the $x$-direction and the $y$-direction. The signal intensity between the samples increased with decreasing distance, but a local minimum could be observed in the profiles for both directions for a distance of 6 mm. The 5 mm distance could still be resolved in $x$-direction, while it failed to be resolved in $y$-direction. In the $z$-direction the resolution is less. A local minimum can be found at 26 mm while a separation fails at 24 mm. The lower resolution can be explained by the lower gradient, as well as the missing excitation and receive chain in $z$-direction. At an iron dose of 125 µg$_{Fe}$ the resolution in the $x$-direction and $y$-direction remained below 10 mm.

**Volumetric imaging sequence**. To image 3D objects the scanner can adjust the height of the sampling plane by moving the selection-field coils up and down using a servo motor. To demonstrate this feature, we prepared a 3D phantom consisting of three slices containing different letters. Each letter was built using glass capillaries with an inner diameter of 1.3 mm filled with diluted Perimag ($c_{Fe} = 8.5$ mg$_{Fe}$ ml$^{-1}$). The slice containing the letter K is placed in the central plane of the imager. The letter U is placed 32 mm above, whereas the letter E is placed 32 mm below the central imaging plane. The scanner needs approximately 0.5 s to move to the yoke (depending on the moved distance). Thirteen slices were measured, with 3 frames per position to avoid motion artifacts. Thus, the total measurement time for the three slices was 26 s. Reconstruction results for three slices of the 3D phantom are shown in Fig. 4. In the images the three letters in the different imaging planes can be identified.

**Static brain experiment**. One target application of the presented system is the detection of an ischemic stroke, which can be identified by a perfusion deficiency in the respective brain area. The stroke is caused by a blood clot or a stenosis in a vessel blocking the blood flow, and causing an under-perfusion of the brain area supplied by the blocked vessel. To prove that MPI is capable of imaging perfusion deficits, a brain phantom was constructed consisting of hollow compartments, which can be filled with water. The right hemisphere contains a volume of 490 ml. The left hemisphere was build with a stroke-like geometry cut out of the phantom leaving a volume of 440 ml.

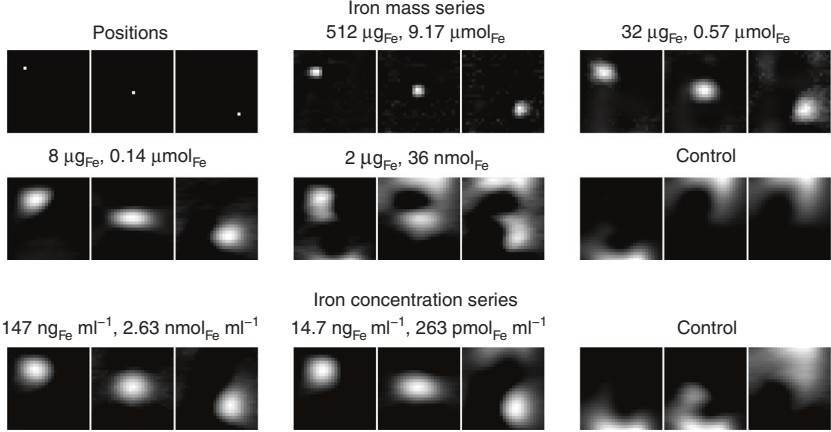

**Fig. 2** Sensitivity of the developed MPI brain scanner. 50 µl samples of Perimag with varying dilutions are moved to three positions on the FOV diagonal (upper 6 images). The sample is considered to be detected if the movement of the sample correlates to the signal shift in the image. Below 2 µg$_{Fe}$ this correlation fails. The same procedure is done with an ellipsoid of 134 ml filled with different concentrations (lower 3 images). At a concentration of 14.7 ng$_{Fe}$ ml$^{-1}$ (263 pmol $_{Fe}$ ml$^{-1}$, 2 µg$_{Fe}$ total) the position is still detectable while it fails at concentrations below that value

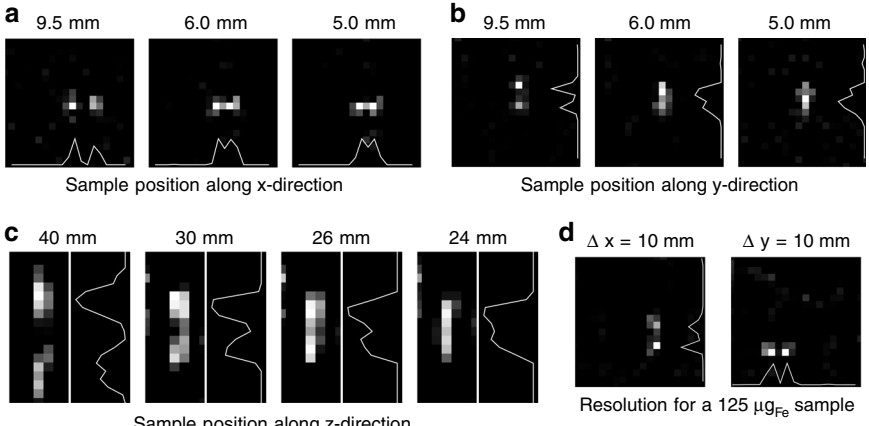

**Fig. 3** Spatial resolution of the developed MPI brain scanner. **a–c** Two samples of 250 µl Perimag with a concentration of 8.5 mg$_{Fe}$ ml$^{-1}$ were placed at varying distance along the $x$—direction, $y$—direction, and $z$-direction. The pixel spacing of the images is 5 mm and marks the lower boundary that can be reached. In $x$-direction the samples can be separated even for a distance of 5 mm. In $y$-direction the samples with 5 mm distance cannot be resolved such that the resolution limit in $y$-direction is 6 mm. In $z$-direction the samples can be resolved down to a distance of 26 mm. **d** For an iron mass of 125 µg$_{Fe}$ the resolution in $x$-direction and $y$-direction remains below 10 mm

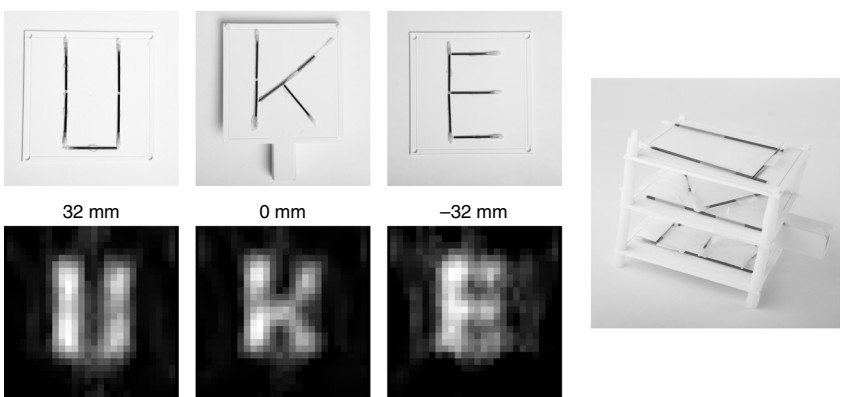

**Fig. 4** Reconstruction results of the 3D phantom. The space covered by the three letters is 90 × 90 mm$^2$ for each layer. The distance between the individual layers is 32 mm. The letters consist of capillaries with an inner diameter of 1.3 mm filled with Perimag with a concentration of 8.5 mg$_{Fe}$ ml$^{-1}$. The reconstructed images show the three letters without interference of the other layers. The total image acquisition time of the 13 slices was 26 s

A fitting part of 42 ml was build which reflects a typical stroke in the MCA-territory[27]. The parts can be filled with varying iron concentrations simulating different grades of perfusion deficits. Both volumes are separated by a total wall thickness of 3 mm. The two hemispheres were placed in the scanner and filled with a concentration of $965\,ng_{Fe}\,ml^{-1}$ ($17.3\,nmol_{Fe}\,ml^{-1}$) Perimag (total iron mass of $938.8\,\mu g$). The stroke part was built four times each filled with a different concentration of $0\,ng_{Fe}\,ml^{-1}$, $318.5\,ng_{Fe}\,ml^{-1}$, $637\,ng_{Fe}\,ml^{-1}$, and $965\,ng_{Fe}\,ml^{-1}$, respectively. The experiments were performed using a 2D imaging sequence. The experiment is considered successful if the stroke region shows a lower concentration than the control experiment, where we inserted the $965\,ng_{Fe}\,ml^{-1}$ concentration.

The results of the static stroke experiments are shown in Fig. 5. For a full perfusion defect (100% reduction) one can identify the dark stroke area on the right side of the image, which has a triangular shape when considering the transversal cross-section through the phantom. Even in the case of a 66% reduction one can see that the particle concentration is lower in the stroke area. For the 33% reduction, the result is less clear. However, in comparison with the control measurement, concentration differences can still be observed in the stroke area. In all cases, a slightly inhomogeneous distribution of the MPI signal can be observed in the left hemisphere with overshoots at the edges, which is an artifact of the Tikhonov regularization technique used in the reconstruction.

Within a monitoring scenario aimed at identifying restenosis, a control image could be made after treatment, when the patient does not suffer from an acute perfusion defect. Using these control measurements, difference images after each monitoring scan can be calculated. The results of this scenario are shown in Fig. 5 as well. In the difference images it is easier to see the lower concentration, in this case as a white area. As the phantom was taken out of the scanner between consecutive measurements the exact position of the phantom inside the bore differed slightly, which leads to subtraction artifacts within the difference images and at the edges of the phantom. This would also happen in a real monitoring scenario when the patient moves the head. Using rigid image registration algorithms, these motion artifacts can be reduced. In an acute case where no control image is available, difference images of the two hemispheres can be calculated.

**Dynamic perfusion experiment**. To evaluate the performance of the scanner in a dynamic setting, we developed a perfusion phantom and performed flow experiments with bolus injections. The flow phantom shown in Fig. 6 consists of two 50 ml tubes filled with 1 mm glass spheres to simulate capillaries within the tissue. The measured remaining fluid volume is 20 ml corresponding to a fill factor of 60%. Each tube represents a hemisphere of a brain. Inside both tubes, a 3D printed part is inserted consisting of two hollow rods, which are connected via tubes to a peristaltic pump delivering an average flow of $100\,ml\,min^{-1}$. The two rods each have a line of evenly distributed holes facing in opposite directions towards the outside of the tube. In this way, the entire length of the tube is perfused homogeneously whereas it is expected that the flow exits the lumen facing towards the center of the phantom. It then passes the glass spheres on its way to the outer side of the phantom and enters the second rod before leaving the phantom towards the pump (see Fig. 6 top right). Four experiments were performed using a 2D imaging sequence, each time injecting a volume of $100\,\mu l$ as fast as possible: two times with a concentration of $c_{Fe} = 8.5\,mg_{Fe}\,ml^{-1}$ ($150\,\mu mol_{Fe}\,ml^{-1}$) and two times with a

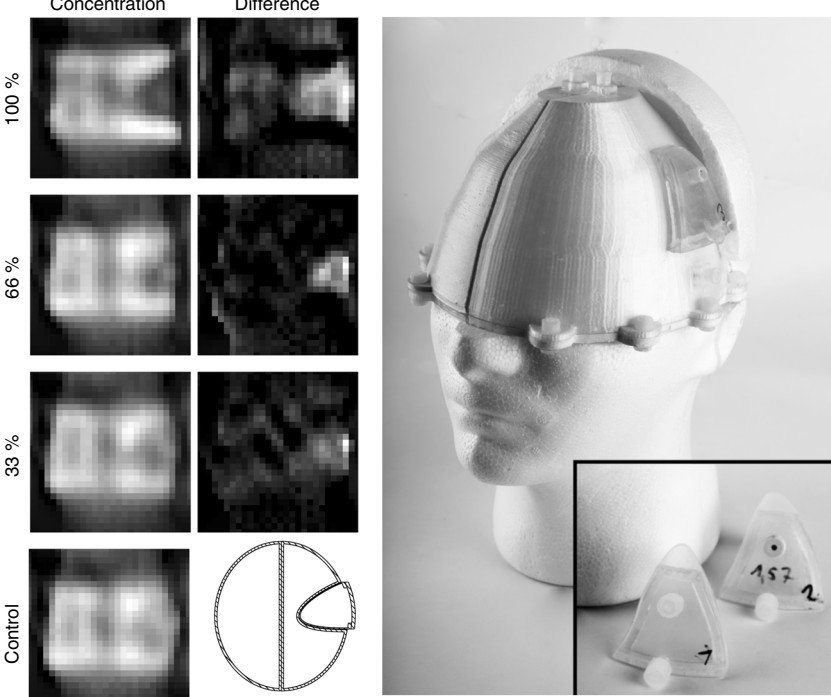

**Fig. 5** Results of the static stroke experiment. (left) Reconstruction results. In the control case both hemispheres and the stroke part were filled with a identical iron concentration of $965\,ng_{Fe}\,ml^{-1}$ with a total amount of $938.8\,\mu g_{Fe}$. For the other cases the iron concentration in the stroke parts was less, which can be identified in the images as a lower particle concentration in all three cases. The stroke parts can be seen in the picture next to the head. (center). The lower concentration in the stroke area becomes even more prominent when subtracting the control image from the stroke images as can be seen in the second column of the images. As reference a cut of the CAD model is drawn next to the control image. (right) Photograph of the phantom mounted on a human head model

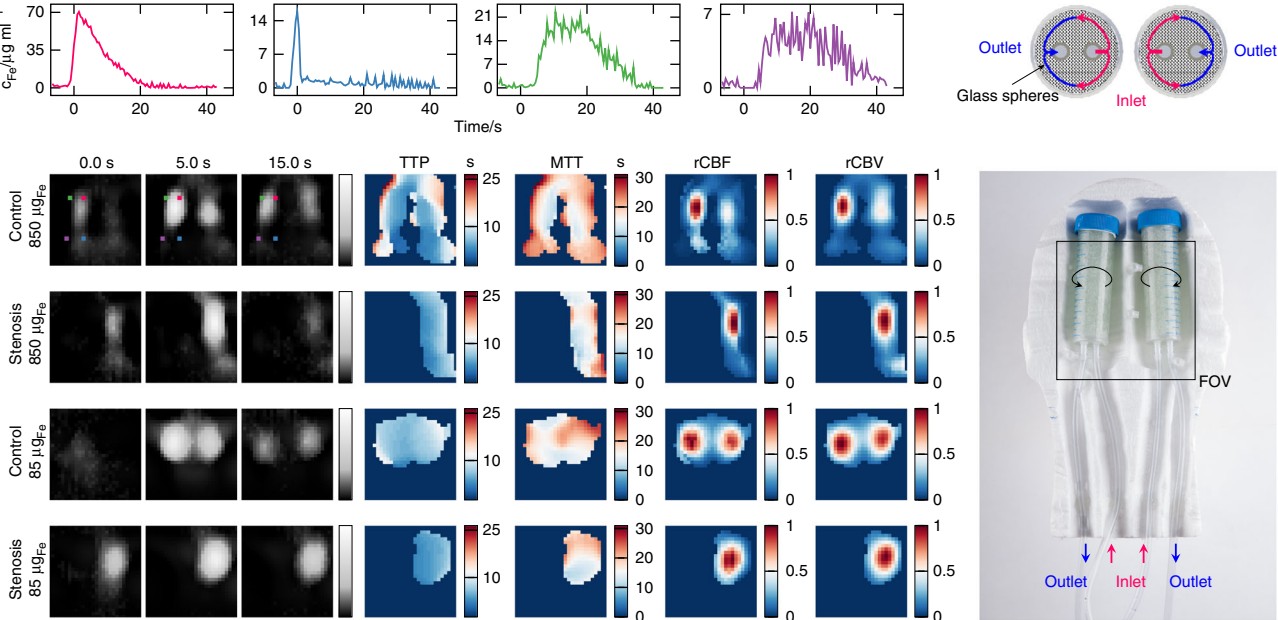

**Fig. 6** Dynamic imaging capabilities of the brain imager. The complete phantom can be seen in the picture on the right. It consists of two 50 ml tubes filled with glass spheres. To visualize the flow that is expected from the phantom a cross section sketch of the tubes is shown above the image. The flow enters the tubes and is evenly distributed via perforated rods inside. The flow exits the rods facing towards the center of the phantom and enters the drain rod facing the outer side of the phantom. In the control experiment the hoses remain untouched while in the stenosis setting the left inflow was pinched. The reconstructed images can be seen in gray scale on the left side. At the lower dosage of 85 µg$_{Fe}$ both tubes were imaged and a stroke setting was detected. With the higher dosage of 850 µg$_{Fe}$ the spatial resolution improved and the inflow and outflow became visible. For the control setting of the experiments with higher concentration, four regions with characteristic temporal behavior were selected and their temporal progression is shown in the graphs (top). The corresponding positions are marked in the reconstructed images of the high dose control experiment. The signal was first detected in the feeding hose (blue graph) with a lower duration compared to the perforated hollow rods (magenta graph). With a slightly higher temporal delay the signal increased in the tubes (green graph) and the discharge hose (purple graph). As the bolus diluted while passing through the phantom, the shape broadens and the signal duration was prolonged. From this time data, the time-to-peak (TTP), mean-transit-time (MTT), relative cerebral-blood-flow (rCBF), and relative cerebral-blood-volume (rCBV) perfusion maps were calculated. The rCBF and rCBV were normalized to the maximum value in the imaging volume. All time data were normalized to the arrival of the bolus

concentration of $c_{Fe} = 0.85$ mg$_{Fe}$ ml$^{-1}$ (15 µmol$_{Fe}$ ml$^{-1}$) which corresponds to a total iron mass of 850 µg and 85 µg, respectively. The bolus was injected at a distance of approximately 1.2 m allowing a dilution on the way to the phantom similar to the dilution within a patient. For both concentrations two different setups were examined. In one case the delivering hoses remained untouched, simulating a healthy brain. In the other case, the hose of the left hemisphere was pinched off, simulating a stroke.

The results of the dynamic perfusion experiment are summarized in Fig. 6 and the Supplementary Videos 1–4. Shown are three selected time points where the bolus passed through the perfusion phantom. For both concentrations the two hemispheres could be identified and spatially resolved. The scanning sequence is also fast enough to track the inflow and the outflow of the tracer material. The images for the 850 µg$_{Fe}$ bolus had a slightly better spatial resolution than the images for the 85 µg$_{Fe}$ bolus. The time data of four selected points are shown in the graphs on top of the Fig. 6. The temporal characteristics are further summarized in the pixel-wise calculated parameter maps (time-to-peak (TTP), mean-transit-time (MTT), relative cerebral-blood-flow (rCBF) and relative cerebral-blood-volume (rCBV)). In the TTP map the lowest values were present in the feeding hose, followed by the rods, the tubes, and the discharge hose. The transit times were summarized in the MTT maps indicating lower values for the inner part of the tubes compared to the outer part. The flow summarized in the rCBF maps was higher in the inner part compared to the outer parts. The integrated particle concentration over time is shown in the rCBV maps.

## Discussion

Within this work, we built a human-scale magnetic particle imaging system tailored for cerebral imaging. The system has a high temporal resolution of 2 frames/s and can image down to an iron mass of 2 µg$_{Fe}$ or an iron concentration of 14.7 ng$_{Fe}$ ml$^{-1}$ (263 pmol$_{Fe}$ ml$^{-1}$, 2 µg$_{Fe}$/134 ml). At high iron concentrations (4.2 mg$_{Fe}$ total), the spatial resolution is in the order of 6 mm within the $xy$-plane and 28 mm in the $z$-direction. With lower concentrations the resolution gets slightly worse. At a total dose of 125 µg$_{Fe}$ the resolution was still above 1 cm. As the system does not make any demands on the building and due to its small size it can be set up as a mobile unit. This allows the diagnosis at the patient's bed instead of bringing the patient to the system.

In practice, it is not possible to obtain the best values for spatial resolution, temporal resolution, sensitivity, and spatial coverage at the same time. In particular, for low iron concentrations, the spatial resolution will be lower as shown in Fig. 2. To evaluate the scanner performance in a more realistic setting, we performed experiments with two different kinds of brain phantoms. The first had an anatomic shape and simulated different degrees of per-fusion deficits. The second was a dynamic flow phantom and demonstrated the capability of the system of resolving dynamic concentration changes. The results show that the scanner is capable of detecting a 42 ml stroke even if the perfusion is reduced by only 33.3%. The dynamic experiments show that it is possible to derive brain perfusion parameter like TTP, MTT, rCBV, and rCBF, which are all very useful for the diagnosis of different brain diseases. One limitation of the dynamic phantom

was the flow (100 ml min$^{-1}$), which was lower than the typical cerebral blood flow (750 ml min$^{-1}$). Nevertheless, the resulting transit times between 10 and 30 s are only slightly above the typical cerebral transit times, which are below 7 s[28] for healthy brain areas and above 7 s for stroke brain areas. We would like to point out that the scanner itself with a frame rate of 2 frames/s would be capable of resolving faster blood flow.

The main use case of MPI scanner presented here is the monitoring of cerebral perfusion within the stroke and intensive care unit. The device could address two different application scenarios. First, the monitoring of treated stroke patients who have a high risk of restenosis, and second, the monitoring of treated hemorrhage patients that have the risk of a rebleed. In both cases, one has to develop a suitable clinical protocol to ensure a regular tracking of the patient's health. The scanner itself has no limitations in application time since it is free of ionizing radiation. However, the tracer material is, of course, limited in its dose that may safely be administered. The approved tracer dose for a 70 kg human for different SPIOs varies between 36 mg$_{Fe}$ (Resovist) and 510 mg$_{Fe}$ (Feraheme)[5,29], and depends on the application the tracer is designed for. These amounts of iron must be compared to about 4–5 g of iron that are typically administered within the human body. However, the usage of these tracers for MPI is an off-label use as the contrast agents have to be approved for a specific procedure. With no clinical MPI imager available, this was not yet possible.

In the following we will consider a dose limitation of 200 mg$_{Fe}$ (2.5 mg$_{Fe}$ kg$^{-1}$ for a 80 kg person[30]). To ensure longtime monitoring, the blood circulation time of the particles is an important factor. Dextran-coated tracers like Resovist usually have a short circulation time below 5 min and accumulate in the liver[31]. Using such a tracer for monitoring would require regularly applying a small bolus. Compared with a single-shot injection of 200 mg$_{Fe}$, the amount of iron in each injection would need to be much smaller, such that the total dosage applied over the course of the monitoring period remains constant. Considering a scenario with a bolus being injected every hour within 72 h, the iron mass needs to be lowered to 2.8 mg$_{Fe}$. Since the blood volume delivered to the brain is approximately 15% of the total cardiac output, a mass of 420 µg iron would pass through the brain, per injection. This is only a factor of 2.2 below the 938.8 µg iron used in our static phantom experiments and a factor of 5 above the 85 µg iron applied in the dynamic phantom experiments. Thus, we can conclude that our scanner would already address this clinical scenario and allow the measurement of brain perfusion parameter maps over a time frame of 72 h. Using long circulating tracers[32] or encapsulation of the tracer into red blood cells[33], it might be either possible to reduce the number of tracer applications or to reduce the amount of tracer that is applied each time.

The next step needs to be the evaluation of the system on living subjects. Prior to this evaluation various safety aspects of the system need to be tested. There are two important risks that need to be addressed. The first is the risk of electric breakdowns into the subject, which can be handled by proper isolation of the drive-field and receive coils. The second risk is the potential of neural nerve stimulation. While we estimated 6 mT $\mu_0^{-1}$ drive-field amplitude to be safe in human subjects, nerve stimulation studies will be necessary to find out the actual nerve stimulation threshold for the head. Concerning clinical application the level of disturbing signals coupling into the receive coils from external electronic systems such as other patient monitoring systems needs to be investigated. However, the system demonstrated to work in unshielded environments.

The scanner characteristic is already sufficient to address the requirements of the clinical application. Nevertheless, there is still potential for further improvement. One key improvement will be

the introduction of a second drive-field coil with field generation in $z$-direction. With that it will be possible to substantially improve the temporal resolution for 3D imaging in the order of 2 frames/s for a volumetric image of about $100 \times 140 \times 100$ mm$^3$. The second drive-field channel combined with a third receive channel in $z$-direction will furthermore substantially improve the spatial resolution in $z$-direction. It is expected to be in the same range as the spatial resolution in $x$-direction.

For even longer surveillance times beyond 72 h or higher resolved images, the sensitivity of the system needs to be improved further. Based on experiments and scaling factors derived in[7] an optimization of the receive coil noise, filter characteristic and the low-noise-amplifier matching should result in a signal-to-noise gain of about a factor of 10. Another factor between 2 and 3 can be gained by using better tracer material[34]. Thus, in total a factor of 20–30 could be reached. This would not only lower the detection limit, but in general also improve the spatial resolution for low tracer dosages.

Today, CBF is monitored using different radiological techniques, like positron emission tomography (PET), single-photon emission computed tomography (SPECT), xenon computed tomography, and brain perfusion imaging with CT or MRI. Except for the xenon-computed tomography, which is not commercially available, none of the other methods can be used at the bedside.

The overall ambition of the imager presented here is the development of a comprehensive and affordable solution for a continuous bedside, imaging-based monitoring of the cerebrovascular status with MPI. The early detection of critical, but treatable incidents, through continuous non-invasive monitoring by MPI, will lead to earlier therapy decision-making in case of a cerebrovascular event and contribute to reduced patient mortality and morbidity. Moreover, monitoring stroke unit patients more continuously and non-invasively will provide a method to reduce the number of CT or MRI scans and lead to a decrease in patient transports, which in turn will reduce patient risks and workload for the medical staff.

## Methods

**Drive-field**. The drive-field generator has a single channel that is directed in $x$-direction (caudal-cranial). The solenoid-like drive-field coil is made of high-frequency litz wire with 1000 strands of 100 µm. The cross section consists of two half circles connected by straight parts resulting in open distances of 190 mm and 230 mm in $y$-direction and $z$-direction, respectively, such that a human head fits into the coil. The power dissipation of the coil is 66 W for generating 6 mT $\mu_0^{-1}$ along the $x$-direction. It does not need additional cooling to thermal convection during measurements. The coil is driven by a power amplifier (AETechron 2105) via a 5th order resonant band-pass filter and a capacitive impedance matching. The drive-field amplitude is controlled using an inductive feedback to the signal generator. The system works with a system clock of 125 MHz. The analog digital converter (ADC) clock is derived from this system clock by a divider of 64 leading to a sampling frequency of $f_{ADC} \approx 1.953$ MHz. The frequency of the drive-field is chosen to be $f_{ADC}/76 \approx 25.699$ kHz resulting in a drive-field period length of 38.912 µs.

**Dynamic selection-field**. To generate the selection-field electromagnetically without cooling, two coils in a Maxwell arrangement are wound around a soft iron yoke that amplifies the magnetic field, provides thermal capacitance and serves as structural support. The coils consist of two copper bands wound around the yoke fixed with epoxy resin. The distance between the coils is 300 mm. Each coil has 648 turns resulting in an inductance of 200 mH and a resistance of 1.8 Ω. Each coil is fed by a programmable current source (SM-1500, Delta Electronica). The currents in the coils can be adjusted individually such that an FFP can be generated at various positions along the $y$-axis by varying the current on both coils. When moving the FFP along a line of 14 cm each coil has a mean power loss of 174 W. Details on the applied imaging sequence are outlined in section 'Imaging sequence'.

**Imaging sequence**. The superposition of the drive- and the selection-field leads to an FFP oscillating along a line in $x$-direction. The particle response has a unique amplitude and phase spectrum in each position along this sampling line since the selection-field provides a unique offset at each point. With a static selection-field

and a 1D excitation one could just measure the 1D particle distribution along the sampling line. In order to cover a 2D slice, we apply sinusoidal shaped currents with a frequency of 2 Hz on each selection-field coil. The phase difference between both fields is $\pi$ such that one coil generates maximum field while the opposing coil generates minimum field. This changing relation of the currents shifts the FFP towards the coil carrying the lower current. In combination with the drive-field the FFP travels along a Cartesian like trajectory[35,36]. The size of the trajectory is about 10 cm in $x$-direction and about 14 cm in $y$-direction. Due to field imperfections, the gradient strength within the FFP is not constant but varies between $0.164\,\mathrm{Tm^{-1}}\,\mu_0^{-1}$ and $0.214\,\mathrm{Tm^{-1}}\,\mu_0^{-1}$. In total one selection-field cycle corresponds to 13,000 drive-field cycles. For data reduction a block-averaging with a factor of 100 is applied such that in total 130 line scans are available.

The yoke handling the selection-field coils is mounted on a linear axis and a servo motor which can move the selection field generator in $x$-direction and $z$-direction. It can be used in two different ways. Either it is used for slice selection in which case the imaging is performed in 2D with high temporal resolution of 2 frames/s . Alternatively, it can be used to form a 3D image by moving the slice up and down and collecting individual slices. This is similar to MRI where volume images can be measured slice by slice as well. For the volumetric measurements and the examination of the spatial resolution in $z$-direction, the yoke was moved from $-30$ mm until 30 mm in 5 mm steps. Within 60 mm a repositioning of the yoke is completed within 0.5 s. Thus, the achievable time resolution of the 3D volumetric images is dependent on the slice thickness of the sequence.

**Receive chain**. When using a 1D excitation with only a collinear receive coil, the system would suffer from a very low spatial resolution in the orthogonal directions caused by the 1D point spread function (PSF) being sharper in the co-linear direction[37]. One way to improve the spatial resolution in the orthogonal direction is to use a multi-dimensional receiver as proposed in ref. [38]. In our system, the $x$-receive coil is realized as a gradiometer coil with 16 receive turns and 22 cancellation turns, and which suppresses background signals from the environment and the direct feed-through of the transmission chain[39]. The $y$-receive coil is realized in a non-gradiometric way, but it is decoupled from the drive-field by orthogonality. The signal is fed to a custom JFET-based low noise amplifier. The amplified signals are digitized with a custom data acquisition system (based on two RedPitaya boards) at a sampling rate of 1.953125 MSs$^{-1}$. The signals are transferred to the measurement PC using Ethernet for storage and reconstruction. All MPI data measured with the system are stored in the magnetic particle imaging data format (MDF)[40].

**Tracer**. All experiments were performed with Perimag (micromod, lot 05617 102-05), a commercially available tracer showing good signal performance similar to that of Resovist[41]. The latter has clinical approval for liver diagnostic in some countries, but it has also been used for brain applications[42]. The undiluted iron concentration of the tracer was $c_{Fe} = 17\,\mathrm{mg_{Fe}\,ml^{-1}}$ (303 $\mu\mathrm{mol_{Fe}\,ml^{-1}}$).

**System calibration and image reconstruction**. To reconstruct images from the measured raw data, prior knowledge of the system and particle behavior is collected by measuring a system matrix[21]. A cubic delta sample (edge length 6.3 mm) containing 250 μl Perimag (micromod, lot 05617 102-05) with an iron concentration of 17 $\mathrm{mg_{Fe}\,ml^{-1}}$ (303.57 $\mu\mathrm{mol_{Fe}\,ml^{-1}}$) is moved on a 2D grid of size $28 \times 28$ covering a FOV of $140 \times 140\,\mathrm{mm}^2$ with a robot while applying a full 2D imaging sequence at each position. The resulting data forms the system matrix and contains the 1D system response for each position of the 2D grid and for each of the 130 patches along the $y$-direction (see Fig. 1). The 2D system matrix is also used for those experiments where the yoke was moved up and down. For the resolution analysis of the system in $z$-direction we measured a second system matrix in the $xz$-plane having a grid of size $7 \times 17$ covering a FOV of $35 \times 85\,\mathrm{mm}^2$. Prior to reconstruction we performed background subtraction for both the system matrix and the object measurement[43,44]. Background data is measured with an empty scanner bore.

For image reconstruction we use a joint approach[45] where the individual 1D measurements are combined prior to reconstruction resulting in a single image for one cycle of the selection-field. The linear imaging equation is solved iteratively with a regularized variant of the Kaczmarz algorithm. The number of iterations is varied from 1 to 500 while the relative regularization parameter[46] is chosen in the range of 0.1 to 100. Both parameters are selected based on the signal strength of the raw data signal. For low signal strength a high regularization parameter and a low number of iterations were used while the spatial resolution experiments having high signal strength were reconstructed with a low regularization parameter and a high number of iterations.

The perfusion parameter maps were calculated as described in the following based on the low-pass filtered pixel-wise concentration-time series. The time-to-peak values were extracted from the series as the time of maximum concentration values. The MTT is the first moment of the time-concentration series[47] divided by the integral of the time series, while the CBV is the integral over time and the CBF is the maximum derivative of the concentration-time series. The CBV and the CBF were normalized to the maximum value in the imaging volume, thus representing relative values rCBV and rCBF.

## Data availability

The authors declare that the data supporting the findings of this study are available within the paper and its supplementary information. Any remaining relevant data and the raw data of the reconstructed images are available from the corresponding author upon reasonable request.

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

## Acknowledgements
We thankfully acknowledge the financial support by the German Research Foundation (DFG, grant number KN 1108/2-1) and the Federal Ministry of Education and Research (BMBF, grant numbers 05M16GKA, 13XP5060B, EuroNanoMed III MAGneTISe, T.K, P.L.). We thank MI Partners Eindhoven for technical assistance and the Institute for Robotics and Cognitive Systems Lübeck for 3D printing the coil frame. We would like to thank Fiona Bailey for the linguistic correction of the article.

## Author contributions
M.G., F.T., P.L., O.We., B.G., and T.K. contributed to critical discussions towards system design and application scenarios. B.G., O.Wo., O.We., and D.v.V. contributed to the design of the dynamic selection- field generator. M.G. and F.T. designed and constructed the remaining MPI components. M.B. and M.M. were modeling the selection fields as basis for sequence programming. T.K. designed and developed the system software. P.S. and T.K. programmed the imaging sequences. F.G. programmed the robotic devices. M.G., F.T., P.S., F.W., N.G., and T.K. designed, planned and performed the imaging experiments. M.G. and T.K. coordinated the project. M.G., T.K., N.G., and P.S. contributed writing the paper. All authors reviewed the manuscript.

## Additional information

**Competing interests:** The authors declare no competing interests.

