## [Peer Review File · Nature Communications]

Reviewers' comments:

Reviewer #1 (Remarks to the Author):

Great piece of work in the MPI field. On the world-wide field of MPI this is top class and certainly novel. This is a big step in the dev. of MPI towards a clinical application in humans. These results and the choice of the application and pathology, is potentially very interesting. In the WW MPI field this paper does certainly show an impressive result.

As the human scanner is in the design and prototype phase, there are presently no results possible on humans. The chosen direction of a head-scanner for this specific pathology does make sense to me.

The main result is maybe presented and written more as a plan to get funding, than to present a new experimental/scientific result.

Reviewer #2 (Remarks to the Author):

The manuscript "Human-sized Magnetic Particle Imaging for Brain Applications" presents the first operational magnetic particle imaging (MPI) device for a possible clinical application. Its intended purpose is for continuous monitoring of stroke patients in ICU units. To build a system that can be used with low demands on space and shielding the authors decided not to move for maximum performance but rather for required performance for the task. Being the first such scanner world-wide this manuscript will be of high importance for other researchers in this field. The research was carried out with great diligence and the manuscript is very well written.

There are a few issues I would raise prior to a possible publication in Nature Communications:

Regarding the content:

- page 2, line 52: for interventional application one could also cite a more recent work that will also increase the number of groups encompassed in the state of the art: Magnetic Particle Imaging Guided Real-Time Percutaneous Transluminal Angioplasty in a Phantom Model. Herz, Stefan; Vogel,

Patrick; Dietrich, Philipp; Kampf, Thomas; Rückert, Martin A.; Kickuth, Ralph; Behr, Volker C.; Bley, Thorsten A. in Cardiovasc. Intervent. Radiol. (2018). 41(7) 1100-1105.

- chapter 2.2: you are claiming the 'lowest iron concentration imaged by MPI so far' but you are performing the measurement in a (compared to others) huge sample. If you multiply the 14.7 ng/ml with the 134 ml of the sample you are back at the 2 ug you already determined before to be the lowest total amount of iron to be imaged. I do not think reducing the detectable concentrations by increasing sample sizes is a valid way to claim 'lowest concentration'.

Please instead of this claim put the concentration into relation to the total amount of iron as I sketched out.

- chapter 2.3 (page 5, line 123): for determining the resolution of the system you are using a c(Fe) of 8.5 mg/ml. This seems a rather high concentration for the desired application. Please comment. If it indeed is higher, you should verify the resolution at a concentration comparable to the real application. Otherwise, higher concentration will result in an over-estimate of resolution (as you point out yourself e.g. page 9, line 200ff.)

- chapter 2.5: for the static brain experiment: were the hemispheres entirely hollow (i.e. completely filled with tracer material?) or did they contain some "pseudo-anatomic" substructure? Please clarify.

- figure 5: please comment on the inhomogeneous signal even in the right hemisphere without a stroke-phantom.

- chapter 2.6: you are examining a 100% stenosis by fully cutting of the pump from one of the hemispheres. Is for a real scenario a partial stenosis of no concern? If it is why did you move to this extreme setting?

- chapter 3, first paragraph: see my comments about imaging low concentration above - I would prefer not to give a concentration without any hint on the volume and therefore the total amount of iron.

- chapter 3, first paragraph (continued): You discuss the resolution at 'higher iron concentrations' - please be quantitative and consider my comment about suitable concentrations for determining the resolution above.

- figure 6: for the sake of being able to better discern details I would prefer the right hand side of fig. 6 (the phantom) to be larger. Maybe split fig. 6 in two figures (phantom and results).

- chapter 3, page 11, first paragraph: why did you not apply higher flow rates if you knew that physiologically these were more relevant?

- chapter 3, page 12, 2nd paragraph and chapter 4.5: since you are discussing steps for a possible clinical trial, please also comment on the availability of human-approved tracer material. You state that Resovist is approved but essentially no longer available in Germany. Are there alternatives?

- chapter 4.1: why is the drive field frequency chosen to 1.953125 MHz / 76? Where does the 76 come from and where the 1.9... MHz? Please elaborate.

Editorial comments:

- page 1, line 6: omit 'the' before brain and diagnosis

- page 2, line 44: allow (not 'allows')

- page 3, line 84: a field-free-point (not 'an')

- page 3, line 90: linearly (not 'linear')

- page 4, figure 1: plot heading: Trajectory (not 'Trajectorie')

- page 4, caption of fig. 1: ...to the left _of_ the imager_

- page 6, caption of fig. 3: (last line) until -> down to

- page 8, caption of fig. 5: ...with a constant iron concentration... -> ...with an identical iron concentration...

Very good work!

Your reviewer Volker C. Behr, University of Würzburg

Reviewer #3 (Remarks to the Author):

Abstract:

Determining the brain perfusion is an important task for the diagnosis and treatment of vascular diseases such as occlusions and intracerebral haemorrhage. Even after successful diagnosis and treatment, there is a high risk of restenosis or rebleeding such that patients need intense and frequent attention in the days after treatment. Within this work, we will present a diagnostic tomographic imager that allows access to brain perfusion information quantitatively in short intervals. The imager is the “rst” (sic) functional magnetic particle imaging device for brain imaging on a human-scale. It is highly sensitive and allows for the detection of an iron concentration of 14.7 ng/ml (263 pmol/ml), which is the lowest iron concentration imaged by MPI so far. The imager is self-shielded and can be used in unshielded environments such as intensive care units. In combination with the low technical requirements this opens a whole variety of possible medical applications and would allow monitoring possibilities on the stroke and intensive care units

Comments:

1. What are the major claims of the paper?

This paper reports a substantial technical advance on a novel imaging modality, Magnetic Particle Imaging (MPI). Although MPI has been around for over a decade (since around 2005), it has been limited to small “rodent size” units, functioning largely for “spectroscopic” type detection of “large” super-paramagnetic iron-oxide (SPIO) nanoparticles, with limited capability to create tomographic images. The authors report three major advances:

- (1) a highly sensitive prototype MPI device that allows detection of iron concentrations as low as 263 pmol/ml, capable of
- (2) human-sized head scanning, for
- (3) brain perfusion imaging.

2. Are they novel and will they be of interest to others in the community and the wider field?

These results are novel, and will be of interest to others both in the community and in the wider field. All medical imaging can be viewed as using different forms of energy to detect & diagnose disease – CT uses X-rays, ultrasound uses sound waves, MRI uses magnetic resonance with RF pulses. The feasibility and practical implementation of a new imaging modality - magnetic moments - to image relevant physiological process such as brain perfusion, at a human scale, is indeed novel.

3. If the conclusions are not original, it would be helpful if you could provide relevant references. Is the work convincing, and if not, what further evidence would be required to strengthen the conclusions?

The major claims/conclusions are original, although there is relevant precedent that should be cited. Specifically, I would consider adding the following reference, which preliminarily explored the design requirements for an MPI human scanner capable of performing functional imaging (this is from the same group as reference 13): Design analysis of an MPI human functional brain scanner. Mason EE, Cooley CZ, Cauley SF, Griswold MA, Conolly SM, Wald LL. Int J Magn Part Imaging. 2017;3(1).

4a. On a more subjective note, do you feel that the paper will influence thinking in the field?

Yes, as per comment #2, above. The current state of development of this technology is somewhat analogous, in my view, to the first prototype MRI scanners, or even the first prototype airplane by Wilbur and Orville Wright; it's impossible to know if future devices will be similar in design and function to what the authors propose here, but the "big idea" proof-of-concept provided is important and – in my opinion – is not only likely to influence thinking in the field, but has the potential to have a substantial future impact in ways we might not yet foresee.

4b. On a more specific note, please comment on the aspects related to stroke imaging, potential applicability of this technique in a clinical scenario and what impact it would make.

As noted in comment #4a, above, this paper has strong potential to have a substantial future impact in ways we might not yet foresee. Speculative examples include, but are not limited to: (1) increased sensitivity for task related functional brain imaging (with conceivable utility for diagnosis of psychiatric, movement, or neuro-degenerative disorders, as well as for pre-neurosurgical treatment planning); (2) CBV, perfusion imaging for brain tumor grading, prognosis, treatment planning, & response monitoring; and (3) more sensitive detection of occult metastases outside the brain for a variety of cancers, given several prior reports of MRI-detected SPIO uptake by reticuloendothelial/lymphatic cells (e.g., the authors may consider citing: Harisinghani MG, Barentsz J, Hahn PF, Deserno WM, Tabatabaei S, van de Kaa CH, de la Rosette J, Weissleder R. Noninvasive detection of clinically occult lymph-node metastases in prostate cancer. *N Engl J Med.* 2003 Jun 19;348(25):2491-9]. As the authors state, MPI has the advantage of avoiding both the ionizing radiation associated with CT scanning, and the expense & potential contraindications associated with MRI.

Regarding the authors' claims of specific clinical indications for their MPI technology in acute stroke patients, however, there are currently not compelling unmet needs for brain perfusion imaging in either a portable ambulance or intensive care unit (ICU) monitoring setting. Indeed, the role of perfusion imaging for patient selection for available acute stroke treatments (intravenous thrombolysis and intra-arterial thrombectomy) in different time windows is currently highly controversial, with recent (2018) American Heart Association imaging guidelines having been rescinded and subsequently re-stated. The authors should therefore consider toning down these claims.

5. Please feel free to raise any further questions and concerns about the paper.

- An extensive copy edit is required for both clarity and proper English grammar & usage. The manuscript should be proofread for typographical errors (e.g., "rst" in the abstract should be "first").

- The authors should refrain from making claims of priority, especially if exaggerated (e.g., "this design has not previously been emphasized in the literature", rather than "we're first"). The facts and references should be clear to the reader, who can draw their own conclusions.

- The authors should define abbreviations at first use for reader-friendliness.

6. We would also be grateful if you could comment on the appropriateness and validity of any statistical analysis, as well the ability of a researcher to reproduce the work, given the level of detail provided.

N/A

ANSWERS TO THE REVIEWERS: HUMAN-SIZED MAGNETIC PARTICLE IMAGING SCANNER FOR HEAD APPLICATIONS

GENERAL COMMENTS

We would like to thank the reviewers for the valuable critics, which helped a lot to improve and shape the focus of the paper. In this reply we outline the changes that we made to the manuscript and answer the issues raised by the reviewers point by point.

REVIEWER 1

Great piece of work in the MPI field. On the world-wide field of MPI this is top class and certainly novel. This is a big step in the dev. of MPI towards a clinical application in humans. These results and the choice of the application and pathology, is potentially very interesting. In the WW MPI field this paper does certainly show an impressive result.

As the human scanner is in the design and prototype phase, there are presently no results possible on humans. The choosen direction of a head-scanner for this specific pathology does make sense to me.

The main result is maybe presented and written more as a plan to get funding, than to present a new experimental/scientific result.

Answer: Thank you for your comment. We wanted to give the reader an impression what the device can reach within a clinical setting. Therefore, many statements within the discussion would fit as well into a funding proposal. However, we think it is important to give the reader the opportunity to see the development in the clinical context. The impression of a funding plan is, in our opinion, also emphasised by the focus on the results rather than the methods. The *methods* section is presented in the form of an appendix, which is demanded by the journal guidelines.

REVIEWER 2

The manuscript "Human-sized Magnetic Particle Imaging for Brain Applications" presents the first operational magnetic particle imaging (MPI) device for a possible clinical application. Its intended purpose is for continuous monitoring of stroke patients in ICU units. To build a system that can be used with low demands on space and shielding the authors decided not to move for maximum performance

but rather for required performance for the task. Being the first such scanner world-wide this manuscript will be of high importance for other researchers in this field. The research was carried out with great diligence and the manuscript is very well written. There are a few issues I would raise prior to a possible publication in Nature Communications:

Regarding the content: - page 2, line 52: for interventional application one could also cite a more recent work that will also increase the number of groups encompassed in the state of the art: Magnetic Particle Imaging Guided Real-Time Percutaneous Transluminal Angioplasty in a Phantom Model. Herz, Stefan; Vogel, Patrick; Dietrich, Philipp; Kampf, Thomas; Rückert, Martin A.; Kickuth, Ralph; Behr, Volker C.; Bley, Thorsten A. in *Cardiovasc. Intervent. Radiol.* (2018). 41(7) 1100-1105.

Answer: Thank you for pointing out this publication. We agree that this paper and the demand for low latency within the intervention is of great value and added this citation within the introduction.

- chapter 2.2: you are claiming the 'lowest iron concentration imaged by MPI so far' but you are performing the measurement in a (compared to others) huge sample. If you multiply the 14.7 ng/ml with the 134 ml of the sample you are back at the 2 ug you already determined before to be the lowest total amount of iron to be imaged. I do not think reducing the detectable concentrations by increasing sample sizes is a valid way to claim 'lowest concentration'. Please instead of this claim put the concentration into relation to the total amount of iron as I sketched out.

Answer: You are right that this iron concentration is corresponding to the 2 μg of the sensitivity by mass experiments. However, within a clinical setting the tracer is distributed in a larger area within the organ of interest. E.g. The human brain has approximately 1.2 l in volume. The sensitivity phantom is in turn representative for the considered application. The sensitivity limit in terms of concentration has to be understood as a combination of the sensitivity in terms of iron mass and the volume of the sensitive region around the FFP. This volume scales with decreasing gradient strength. E.g. a system with 2.5 T/m with a sensitivity of 2 $\mu\text{g}(\text{Fe})$ will most probably not be able to image the 286 pmol/ml within the 134 ml volume. One other possibility would be to define the sensitive volume, but currently no good direct approach has been shown to measure it. In addition, lowering the gradient on systems designed for 2.5 T/m leads to rising background signals from unsaturated magnetic material around the bore. At least that is something we see in our pre-clinical imaging system. This is why we think, both values are important measures for the sensitivity. However we added the value of the 2 μg of total iron used in each sentence to avoid a misinterpretation of the values and weakened the statement of the lowest concentration imaged.

- chapter 2.3 (page 5, line 123): for determining the resolution of the system you are using a $c(\text{Fe})$ of 8.5 mg/ml. This seems a rather high concentration for the

desired application. Please comment. If it indeed is higher, you should verify the resolution at a concentration comparable to the real application. Otherwise, higher concentration will result in an over-estimate of resolution (as you point out yourself e.g. page 9, line 200ff.)

Answer: Thank you for that comment. The measurement of the iron concentration based resolution is not easily done. The achievable resolution is not only dependent on the concentration of the sample, but also on the total amount of tracer within the FOV. Therefore, even a measurement of a diluted sample would not resemble the setting within a human. The values shown here must be interpreted more in a technical sense. The proof that the system is able to provide sufficient resolution in an application case are shown in the phantom experiments.

- chapter 2.5: for the static brain experiment: were the hemispheres entirely hollow (i.e. completely filled with tracer material?) or did they contain some "pseudo-anatomic" substructure? Please clarify.

Answer: Thank you for your comment on this ambiguity. There was no substructure within the phantom. It was entirely hollow. We added a sentence to clarify that.

Revised: To prove that MPI is capable of imaging perfusion deficits, a brain phantom was constructed consisting of hollow compartments which can be filled with water.

- figure 5: please comment on the inhomogeneous signal even in the right hemisphere without a stroke-phantom.

Answer: The inhomogeneity of the MPI signal is a usual reconstruction artefact when applying Tikhonov regularisation, which leads to overshoots at the edges. We added a sentence to clarify this. There are possibilities to reduce these kinds of artefacts by applying prior-knowledge based regularisation techniques (e.g. total variation regularisation) but these might also bias the signal in the stroke region and in turn it is not yet clear what the best approach will be for the reconstruction of MPI brain images.

Revised: In all cases, a slightly inhomogeneous distribution of the MPI signal can be observed in the left hemisphere with overshoots at the edges, which is an artifact of the Tikhonov control technique used in the reconstruction.

- chapter 2.6: you are examining a 100% stenosis by fully cutting of the pump from one of the hemispheres. Is for a real scenario a partial stenosis of no concern? If it is why did you move to this extreme setting?

Answer: From the beginning, we were planning all the experiment with a neurologist from our clinic, who is working in the largest stroke center in northern Germany. Although 100% stenosis seems to be extreme, most patients with an acute stroke, especially severely disabled patients, present with complete occlusion of a vessel. This is the reason why we chose this realistic setting. Of course,

partial stenosis is of great interest too, and we will address this topic in further studies.

- chapter 3, first paragraph: see my comments about imaging low concentration above - I would prefer not to give a concentration without any hint on the volume and therefore the total amount of iron.

Answer: We added the missing values (volume and contained iron) to this sentence to clarify.

Revised: The system has a high temporal resolution of 2 frames/s and can image down to an iron mass of $2\ \mu\text{g}(\text{Fe})$ or an iron concentration of $14.7\ \text{ng ml}^{-1}$ ($263\ \text{pmol ml}^{-1}$, $2\ \mu\text{g}(\text{Fe}) / 134\ \text{ml}$).

- chapter 3, first paragraph (continued): You discuss the resolution at 'higher iron concentrations' - please be quantitative and consider my comment about suitable concentrations for determining the resolution above.

Answer: We added a sentence about our observations regarding the iron concentrations of the samples in respect to the achieved resolution.

Revised: At high iron concentrations ($4.2\ \text{mg}(\text{Fe})$ total), the spatial resolution is in the order of 6 mm within the xy -plane and 28 mm in the z -direction. With lower concentrations the resolution gets slightly worse. At a total iron dose of $100\ \mu\text{g}(\text{Fe})$ the resolution was still above 1 cm (not shown).

- figure 6: for the sake of being able to better discern details I would prefer the right hand side of fig. 6 (the phantom) to be larger. Maybe split fig. 6 in two figures (phantom and results).

Answer: We were experimenting with the size and position of the picture and scaled it to appear bigger in the figure. In our opinion, splitting the figure in two makes it harder to connect both figures which is why we would like to include the picture next to the reconstruction results. But in the end the figure placement will undergo editorial review by the Nature journal so that the actual figure positioning may change in the final version of the manuscript.

- chapter 3, page 11, first paragraph: why did you not apply higher flow rates if you knew that physiologically these were more relevant?

Answer: Thank you for this comment. The limited flow was mostly due to the capacity of our pump. However, since the perfusion phantom did also not 100% match the behaviour in the human brain (different volume, no true vasculature) we decided to match the mean transit times (MTT) of our experiment to that observed in the human brain. A higher flow-rate would lead to much shorter MTT which are not physiologically feasible. This shows that we need to develop more realistic brain phantoms, which will be subject of a future work.

- chapter 3, page 12, 2nd paragraph and chapter 4.5: since you are discussing steps for a possible clinical trial, please also comment on the availability of human-approved tracer material. You state that Resovist is approved but essentially no longer available in Germany. Are there alternatives?

Answer: We added a sentence on the usage of clinically approved tracers for MPI. As the tracer has to be approved for its given purpose, every MRI approved tracer would be an off label use. We agree, that the MPI community is in need of a clinical tracer approved for MPI, and we hope that our work might lead to more research in the field of MPI tracers and clinical trials for approval.

Revised: However, the usage of these tracers for MPI is an off-label use as contrast agents have to be approved for a specific procedure. With no clinical MPI imager available, this was not yet possible.

- chapter 4.1: why is the drive field frequency chosen to $1.953125 \text{ MHz} / 76$? Where does the 76 come from and where the 1.9... MHz? Please elaborate.

Answer: The drive field frequency has to be derived from the internal clock of the IO cards (125 MHz). To match the frequency range of common drive fields, this clock has to be divided to lower values. To create a closed trajectory the dividers for all used drive- and focus field frequencies have to be integers. The 1.953125 MHz is caused by a division factor of 64 from the 125 MHz clock frequency. The drive field then is further divided by the 76, thus one drive field cycle contains 76 ADC values. The other frequencies (y and future z axis) are derived in a similar way. We added some sentences to clarify this.

Revised: The system works with a system clock of 125 MHz. The ADC clock is derived from this system clock by a divider of 64 leading to a sampling frequency of $f_{\text{ADC}} \approx 1.953 \text{ MHz}$. The frequency of the drive-field is chosen to be $f_{\text{ADC}} / 76 \approx 25.699 \text{ kHz}$ resulting in a drive-field period length of $38.912 \mu\text{s}$.

Editorial comments: - page 1, line 6: omit 'the' before brain and diagnosis

- page 2, line 44: allow (not 'allows')

- page 3, line 84: a field-free-point (not 'an')

- page 3, line 90: linearly (not 'linear')

- page 4, figure 1: plot heading: Trajectory (not 'Trajectorie')

- page 4, caption of fig. 1: ...to the left of the imager

- page 6, caption of fig. 3: (last line) until $-j$ down to

- page 8, caption of fig. 5: ...with a constant iron concentration... ...with an identical iron concentration...

Answer: We corrected all the editorial comments. Thank you!

Very good work! Your reviewer Volker C. Behr, University of Würzburg

1. REVIEWER 3

Abstract:

Determining the brain perfusion is an important task for the diagnosis and treatment of vascular diseases such as occlusions and intracerebral haemorrhage. Even after successful diagnosis and treatment, there is a high risk of restenosis or re-bleeding such that patients need intense and frequent attention in the days after treatment. Within this work, we will present a diagnostic tomographic imager that allows access to brain perfusion information quantitatively in short intervals. The imager is the “rst” (sic) functional magnetic particle imaging device for brain imaging on a human-scale. It is highly sensitive and allows for the detection of an iron concentration of 14.7 ng/ml (263 pmol/ml), which is the lowest iron concentration imaged by MPI so far. The imager is self-shielded and can be used in unshielded environments such as intensive care units. In combination with the low technical requirements this opens a whole variety of possible medical applications and would allow monitoring possibilities on the stroke and intensive care units

Comments:

1. What are the major claims of the paper?

This paper reports a substantial technical advance on a novel imaging modality, Magnetic Particle Imaging (MPI). Although MPI has been around for over a decade (since around 2005), it has been limited to small “rodent size” units, functioning largely for “spectroscopic” type detection of “large” super-paramagnetic iron-oxide (SPIO) nanoparticles, with limited capability to create tomographic images. The authors report three major advances: (1) a highly sensitive prototype MPI device that allows detection of iron concentrations as low as 263 pmol/ml, capable of (2) human-sized head scanning, for (3) brain perfusion imaging.

2. Are they novel and will they be of interest to others in the community and the wider field?

These results are novel, and will be of interest to others both in the community and in the wider field. All medical imaging can be viewed as using different forms of energy to detect & diagnose disease – CT uses X-rays, ultrasound uses sound waves, MRI uses magnetic resonance with RF pulses. The feasibility and practical implementation of a new imaging modality - magnetic moments - to image relevant physiological process such as brain perfusion, at a human scale, is indeed novel.

Answer: Thank you for your comment. We as well hope that the results will be of interest for the MPI community and for a wider field.

3. If the conclusions are not original, it would be helpful if you could provide relevant references. Is the work convincing, and if not, what further evidence would be required to strengthen the conclusions?

The major claims/conclusions are original, although there is relevant precedent that should be cited. Specifically, I would consider adding the following reference, which preliminarily explored the design requirements for an MPI human scanner capable of performing functional imaging (this is from the same group as reference 13): Design analysis of an MPI human functional brain scanner. Mason EE, Cooley CZ, Cauley SF, Griswold MA, Conolly SM, Wald LL. *Int J Magn Part Imaging*. 2017;3(1).

Answer: We totally agree with the reviewer that this publication was indeed missing in the paper. We actually did include it during the preparation of the manuscript since it is one of the first papers on a dedicated MPI head scanner but it seems that it was removed by mistake. We added the reference back into the introduction and thank the reviewer for this important hint.

Revised: Nevertheless, simulation studies on the design of a functional MPI brain imager proved promising capabilities for human scale systems [23].

4a. On a more subjective note, do you feel that the paper will influence thinking in the field?

Yes, as per comment 2, above. The current state of development of this technology is somewhat analogous, in my view, to the first prototype MRI scanners, or even the first prototype airplane by Wilbur and Orville Wright; it's impossible to know if future devices will be similar in design and function to what the authors propose here, but the "big idea" proof-of-concept provided is important and – in my opinion – is not only likely to influence thinking in the field, but has the potential to have a substantial future impact in ways we might not yet foresee.

Answer: We thank you for this encouraging comment. We ourselves see great potential in this system and hope that, in the future, we can look back and see your comparisons match.

4b. On a more specific note, please comment on the aspects related to stroke imaging, potential applicability of this technique in a clinical scenario and what impact it would make.

As noted in comment 4a, above, this paper has strong potential to have a substantial future impact in ways we might not yet foresee. Speculative examples include, but are not limited to: (1) increased sensitivity for task related functional brain imaging (with conceivable utility for diagnosis of psychiatric, movement, or neurodegenerative disorders, as well as for pre-neurosurgical treatment planning); (2) CBV, perfusion imaging for brain tumor grading, prognosis, treatment planning, & response monitoring; and (3) more sensitive detection of occult metastases outside the brain for a variety of cancers, given several prior reports of MRI-detected SPIO uptake by reticuloendothelial/lymphatic cells (e.g., the authors may consider citing: Harisinghani MG, Barentsz J, Hahn PF, Deserno WM, Tabatabaei S, van de Kaa CH, de la Rosette J, Weissleder R. Noninvasive detection of clinically occult lymph-node metastases in prostate cancer. *N Engl J Med*. 2003 Jun

19;348(25):2491-9]. As the authors state, MPI has the advantage of avoiding both the ionizing radiation associated with CT scanning, and the expense & potential contraindications associated with MRI.

Answer: We added a sentence within the introduction regarding the usage of SPIONs with MRI for the imaging of different diseases. Here, we included the suggested reference and thank the reviewer for the valuable suggestion.

Revised: In MRI such contrast agents are used for various applications including the detection of lymph node metastasis [3], imaging of liver tissue [4] and imaging of intra-abdominal lesions within the bowel [5].

Regarding the authors' claims of specific clinical indications for their MPI technology in acute stroke patients, however, there are currently not compelling unmet needs for brain perfusion imaging in either a portable ambulance or intensive care unit (ICU) monitoring setting. Indeed, the role of perfusion imaging for patient selection for available acute stroke treatments (intravenous thrombolysis and intra-arterial thrombectomy) in different time windows is currently highly controversial, with recent (2018) American Heart Association imaging guidelines having been rescinded and subsequently re-stated. The authors should therefore consider toning down these claims.

Answer: Thank you very much for your comment. We agree that some of our statements that we made were enthusiastic. But we note that before designing the scanner and planning all the experiments, we were discussing the need of an MPI scanner in the clinical setting with our colleagues from the Department of Neurology, who treat stroke patients on a daily basis. Our intention is not to replace CT or MRI, but we see an unmet need for bedside monitoring devices of cerebral perfusion for severely ill patients, whose lives are at risk during every transportation in the clinic (in different clinical setting, e.g., the stroke unit, ICU, intraoperative monitoring). Indeed, a portable MPI scanner in an ambulance is a very futuristic setting, but there are some promising studies about ultra-early detection of stroke with CT scanners in ambulances in Germany and the United States (<https://www.ncbi.nlm.nih.gov/pubmed/28461420>; <https://www.ncbi.nlm.nih.gov/pubmed/26508753>). To clarify the role of an MPI scanner in the clinic, we adjusted the last paragraph accordingly.

Revised: Today, CBF is monitored using different radiological techniques, like positron emission tomography (PET), single-photon emission computed tomography (SPECT), xenon computed tomography, and brain perfusion imaging with CT or MRI. Except for the xenon-computed tomography, which is not commercially available, none of the other methods can be used at the bedside. The overall ambition of the imager presented here is the development of a comprehensive and affordable solution for a continuous bedside, imaging-based monitoring of the cerebrovascular status with MPI. The early detection of critical, but treatable incidents, through continuous non-invasive monitoring by MPI, will lead to earlier therapy decision-making in case of a cerebrovascular event and contribute

to reduced patient mortality and morbidity. Moreover, monitoring stroke unit patients more continuously and non-invasively will provide a method to reduce the number of CT or MRI scans and lead to a decrease in patient transports, which in turn will reduce patient risks and workload for the medical staff.

5. Please feel free to raise any further questions and concerns about the paper.

- An extensive copy edit is required for both clarity and proper English grammar & usage. The manuscript should be proofread for typographical errors (e.g., “rst” in the abstract should be “first”). - The authors should refrain from making claims of priority, especially if exaggerated (e.g., “this design has not previously been emphasized in the literature”, rather than “we’re first”). The facts and references should be clear to the reader, who can draw their own conclusions. - The authors should define abbreviations at first use for reader-friendliness.

Answer: Thank you for the comment. We adapted the sentences in accordance to your suggestions. In terms of English grammar and usage, we gave the publication to a British native speaker for linguistic proof read.

Revised: Abstract: Within this work, we will present a diagnostic tomographic imager that allows access to brain perfusion information quantitatively in short intervals. The imager is the first attempt to implement a magnetic particle imaging device for brain applications on a human-scale.

Revised: Discussion: Within this work, we built a human-scale magnetic particle imaging system tailored for cerebral imaging.

6. We would also be grateful if you could comment on the appropriateness and validity of any statistical analysis, as well the ability of a researcher to reproduce the work, given the level of detail provided.

N/A

REVIEWERS' COMMENTS:

Reviewer #2 (Remarks to the Author):

The manuscript "Human sized Magnetic Particle Imaging for Brain Applications" has been revised to my full satisfaction. As well my own comments as those of my co-reviewers have been - in my view - duly addressed.

I now consider the manuscript fit - and fully suitable - for a publication in Nature Communications.

Congratulations on this excellent piece of work!

Volker C. Behr, University of Würzburg

Reviewer #3 (Remarks to the Author):

The authors appropriately addressed my Reviewer 3 comments and suggestions.